# Determination of d-Cycloserine Impurities in Pharmaceutical Dosage Forms: Comparison of the International Pharmacopoeia HPLC–UV Method and the DOSY NMR Method

**DOI:** 10.3390/molecules25071684

**Published:** 2020-04-07

**Authors:** Damjan Makuc, Živa Švab, Katerina Naumoska, Janez Plavec, Zdenko Časar

**Affiliations:** 1Slovenian NMR Centre, National Institute of Chemistry, Hajdrihova 19, SI-1000 Ljubljana, Slovenia; damjan.makuc@ki.si (D.M.); janez.plavec@fkkt.uni-lj.si (J.P.); 2Faculty of Chemistry and Chemical Technology, University of Ljubljana, Večna pot 113, SI-1001 Ljubljana, Slovenia; ziva.marsetic@icgeb.org; 3Lek Pharmaceuticals d.d., Sandoz Development Center Slovenia, Verovškova ulica 57, SI-1526 Ljubljana, Slovenia; 4Department of Food Chemistry, National Institute of Chemistry, Hajdrihova 19, SI-1000 Ljubljana, Slovenia; katerina.naumoska@ki.si; 5EN-FIST Centre of Excellence, Trg Osvobodilne fronte 13, SI-1000 Ljubljana, Slovenia; 6University of Ljubljana, Faculty of Pharmacy, Aškerčeva cesta 7, SI-1000 Ljubljana, Slovenia

**Keywords:** d-cycloserine, cycloserine dimer, HPLC, NMR, DOSY

## Abstract

d-cycloserine is a broad-spectrum antibiotic that is currently being used as a secondary choice in the treatment of tuberculosis. In recent years, it has become more popular, due to its effect on the nervous system. In this current study, we provide evidence that The International Pharmacopoeia HPLC–UV method for d-cycloserine impurity profiling is not repeatable due to the variable response of cycloserine dimer, one of d-cycloserine impurities. Therefore, we introduced the DOSY (diffusion ordered spectroscopy) NMR (nuclear magnetic resonance) technique to determine the levels of d-cycloserine impurities in pharmaceutical dosage forms. The DOSY NMR technique allowed separation of d-cycloserine, its degradation products, and key process impurities in concentrations below pharmacopoeial specification limits. The proposed DOSY NMR method allowed accurate identification and quantification of the cycloserine dimer, which was not possible through the use of the pharmacopoeial HPLC method. The current method has the potential for practical use in analytical laboratories of the pharmaceutical industry.

## 1. Introduction

d-Cycloserine is a broad-spectrum antibiotic, which can be biosynthesized by *Streptomyces garyphalus*, *Streptomyces orchidaceous*, *and Streptomyces lavendulae*, but can also be obtained by synthesis. It was first isolated in 1955 and in vitro tested against *M. tuberculosis* in 1966 [1]. It is classified as a second-line antitubercular agent for the treatment of multidrug-resistant tuberculosis and is used in combined therapies, along with other antitubercular drugs to bypass its toxic effects (i.e., neurotoxicity) [1]. In addition, it could be potentially used to treat neuropsychological disorders (e.g., schizophrenia, anxiety, major depression, autism, and Alzheimer’s disease), due to its agonistic effect on *N*-methyl-d-aspartate (NMDA) receptors [1,2].

To date, several analytical methods have been reported for the determination of d-cycloserine in different matrices, mostly using high-performance liquid chromatography (HPLC). Different approaches like ion-pair chromatography [3,4,5], aqueous normal phase (ANP) chromatography [6], or hydrophilic interaction liquid chromatography (HILIC) [7] have been applied so far to retain the highly hydrophilic analyte and its analogues, on a column.

Since the d-cycloserine molecule does not contain chromophores, it has been usually derivatized prior to (by *p*-benzoquinone [8] or 9-chloro-10-methyl acrydinium triflate [9]) or after (by *o*-phtalaldehyde [5]) its HPLC column separation and, thus, detected by a fluorescence detector (FLD). However, none of these methods included determination of its main degradation product, cycloserine dimer (3,6-bis(aminooxymethyl)piperazine-2,5-dione, for structure see Figure 1), which does not form a fluorescent product, and consequently, cannot be detected by FLD.

Several issues regarding the HPLC–UV method for the determination of impurities in pharmaceutical dosage forms containing d-cycloserine have been met so far. Particularly, the quantification of the cycloserine dimer has been challenging. An ion-pair HPLC–UV method was proposed to identify and quantify d-cycloserine and its degradation products [4]. The aforementioned work reported unrepeatable cycloserine dimer peak areas and consequently hampered quantification thereof. It was speculated that the reason might lie in the nucleophilic nature of this compound, causing its binding to free silanol groups of the column [4]. To solve the problem, column conditioning through successive injections of the cycloserine dimer in a high concentration of 0.5 mg mL^−1^ to 2.0 mg mL^−1^ was suggested [4]. This method was further optimized [3] and later published in The International Pharmacopoeia (Ph. Int.) [10,11]. The International Pharmacopoeia mentions two known impurities, the d-serine (for structure see Figure 1) and cycloserine dimer [10,11].

Kaushal et al. developed another HPLC–UV method for d-cycloserine impurity profiling [12]. This method showed insufficient resolution between the peaks of d-cycloserine, d-serine, and hydroxylamine, while the dimer was not included in the study [12]. The United States Pharmacopoeia (USP) contains only a method for d-cycloserine assay determination [13]. In addition, it has been reported that the USP method does not provide a stable baseline [3]. Only ANP chromatography coupled with mass spectrometry (MS) showed a successful determination of d-cycloserine and its dimer [6].

Several methods, including HPLC–UV [14], HPLC coupled with mass spectrometry (MS) [15] or tandem mass spectrometry (MS/MS) [7,16,17,18], for the determination of d-cycloserine in human plasma have also been reported. However, these approaches do not consider nor measure cycloserine dimer as an impurity. Similarly, in another study, only cycloserine was determined in microdialysis samples through LC–MS/MS, after derivatization with benzoyl chloride [19]. A (reversed-phase) RP–HPLC method for the determination of enantiomeric purity of cycloserine, using chiral derivatizing reagents, *o*-phthalaldehyde, and *N*-acetyl-cysteine was also proposed [20].

Two studies reported on the dimerization of d-cycloserine initiated by acetonitrile when used as a sample solvent and strongly recommended methanol use instead [6,21]. This dimerization could be further enhanced in the electrospray ionization (ESI) source [21]. In addition, it was found that d-cycloserine is unstable under acidic conditions (pH 1–2) [12,22], converting into its dimer in the solid-state and in solution [22]. Interestingly, a study from 1962 proposed that the active form of d-cycloserine in vivo is actually its dimer [23]. However, it has become imperative to purify cycloserine from the dimer, which is nowadays considered to be a contaminant [23]. Recently, a new study reported on severe peak tailing of a d-cycloserine peak, due to its on-column dimerization, promoted by stationary phase metal oxides, which was resolved by addition of citrate (chelating agent) to the HILIC mobile phase [24].

The present study showed that d-cycloserine impurities could not be reliably quantified in a final dosage form using The International Pharmacopoeia method [10,11]. Therefore, a new approach was proposed using the DOSY NMR technique to determine the levels of its impurities. Few reports exist in the literature, where the quality of different products (e.g., Fluoxetine, Fluvoxamine [25], heroin [26] and libido improving food supplements [27]) was determined by DOSY NMR. To this end, the concentration of the analytes was determined by ^1^H spectra, while using 3-(trimethylsilyl)propionic-2,2,3,3-*d*_4_ acid sodium salt (TMSP) and maleic acid as standards.

## 2. Results and Discussion

### 2.1. Verification of The International Pharmacopoeia (Ph. Int.) HPLC Method for Determination of Impurities in Samples Containing d-Cycloserine

#### 2.1.1. Repeatability of d-Cycloserine Chromatographic Peak Areas

Standard d-cycloserine in concentration of 0.0015 mg mL^−1^ was injected ten times in sequence. Relative standard deviation (RSD) for the area of d-cycloserine peaks obtained from all ten injections was calculated to be 0.58%, which proved the repeatability of the analytical system for determination of d-cycloserine. Moreover, retention times of d-cycloserine were repeatable as well.

#### 2.1.2. Determination of the Limit of Quantification (LOQ) for d-Cycloserine

For determination of LOQ, five different concentrations of d-cycloserine in the range of 0.001% to 0.05%, relative to its concentration in the sample were prepared and the peak signal to noise ratio (S/N) was checked. The concentration at 0.05% (*S*/*N* = 52.5) was determined as LOQ. At this level, the peak was easily integrable, while the concentration was still below the specification limit (no more than (NMT) 0.15% in active pharmaceutical ingredient and NMT 0.4% in final dosage form [10,11]).

#### 2.1.3. Linearity of the d-cycloserine Signal

The solutions for the linearity study were prepared, as shown in Table 1 (Lin1–Lin9), and the linearity of the d-cycloserine signal in the range of 0.05% to 0.53% d-cycloserine, relative to its concentration in the sample was tested. The equation of the curve was y = 91799x + 404.25, where y stands for area and x for concentration of d-cycloserine in µg mL^−1^. R^2^ was 0.9999 (R^2^ ≥ 0.99), which confirmed the linear response of the d-cycloserine signal in the determined range.

#### 2.1.4. Determination of Retention Times of d-Cycloserine and Its Impurities

Retention times for d-cycloserine and its impurities were checked at the wavelength of 219 nm, separately for each compound. Among the evaluated impurities, we considered the cycloserine dimer and d-serine (Figure 1), which are both known process impurities (related substances), as well as the degradation products of the final dosage product from the Ph. Int. monographs [10,11]. In addition, we also examined the 3-chloro-d-alanine methyl ester hydrochloride (for structure see Figure 1), which is a well-known process impurity of d-cycloserine [28,29,30,31]. The solutions were injected with concentrations of about 0.77 mg mL^−1^. The d-serine peak area was significantly lower in comparison to that of d-cycloserine. The peak of the 3-chloro-d-alanine methyl ester hydrochloride was not observed, while the cycloserine dimer gave two chromatographic peaks that have been reported to correspond to its isomeric forms [3]. Depending on the injection number, the symmetry of cycloserine dimer peaks was varied, but in general the peaks were not bell-shaped (Gaussian). Interestingly, when another chromatographic (ANP) method was coupled with mass spectrometric detection was applied, only one peak corresponding to the cycloserine dimer was observed [6].

Since the 3-chloro-d-alanine methyl ester hydrochloride peak was not observed at 219 nm, UV spectra of all compounds that were separately dissolved in mobile phase A, were acquired in the range of 190 nm to 400 nm. Figure 2 clearly shows that the d-cycloserine impurities (cycloserine dimer, d-serine, and 3-chloro-d-alanine methyl ester hydrochloride) absorb at lower wavelengths in comparison to the d-cycloserine molecule, which was most likely the reason why the 3-chloro-d-alanine methyl ester hydrochloride peak was not observed previously and the peak for d-serine was of really low intensity. Although the issue with the d-serine peak could be resolved by including its response factor in the calculations, the tested method [3] was not suitable for the determination of 3-chloro-d-alanine methyl ester hydrochloride. Since d-serine and the 3-chloro-d-alanine methyl ester hydrochloride absorb at lower wavelengths, chromatograms of all compounds were recorded again at 190 nm, despite the noisy baseline and incompatibility of the mobile phase at this wavelength.

Using the Ph. Int. HPLC method, the elution order of the tested compounds starting from the lowest retention time was as follows—3-chloro-d-alanine methyl ester hydrochloride < d-serine < d-cycloserine < cycloserine dimer 1 < cycloserine dimer 2. The cycloserine dimer in the acidic mobile phase (pH 2.8) harbored two positively charged amino groups, which might result in high affinity to the negatively charged groups of sodium 1-octanesulfonate (SOS) in the stationary phase (ion-exchange chromatography) and, thus, the highest retention time. Both, d-cycloserine and d-serine had only one positively charged amino group. The reason for the difference in their retention times lies in their polarity. Namely, d-serine was more polar with a polar surface area of 83.5 Å, in comparison to d-cycloserine with a polar surface area of 64.4 Å. Although 3-Chloro-d-alanine methyl ester hydrochloride has a polar surface area of 52.3 Å, it is present in the form of an ionic salt, which could be the reason for its early elution. Indeed, 3-Chloro-d-alanine methyl ester hydrochloride elutes with the dead volume, which makes its quantification with this method impossible. Data for the polar surface areas of the molecules were obtained with the software ACD/Labs Perceptas Platform [32].

#### 2.1.5. Saturation of Column Free Silanol Groups with Cycloserine Dimer and Repeatability of Cycloserine Dimer Chromatographic Peak Areas

As for d-cycloserine, linearity, repeatability, and other validation parameters should also be tested for its impurities, which are known to be present in the final dosage form. To date, the linearity of the reported HPLC method for the determination of d-cycloserine-related substances [3], which was later adopted as the official method and was included in the Ph. Int., was performed only by evaluating the response of d-cycloserine.

Burge et al. (2005) reported an issue with the repeatability of the cycloserine dimer chromatographic peak areas, speculating that this could be due to the fact that the nucleofilic cycloserine dimer could bind to the free silanol groups of the column [4]. Therefore, it was suggested to saturate the column with cycloserine dimer, before the analyses [4]. Pendela et al. (2008) reported that they have not observed this issue with an Hypersil BDS column [3] and their method was later included in The International Pharmacopoeia [10,11].

The Ph. Int. HPLC method for determination of d-cycloserine-related substances was tested using the new Hypersil BDS column with included column saturation, using two times 50 µL of the cycloserine dimer solution (0.5 mg mL^−1^) [4]. Then, cycloserine dimer in the same concentration and volume was consecutively injected twenty times from the same HPLC vial. The obtained chromatograms of the cycloserine dimer are shown in Figure 3. The first four injections resulted in no or very little response. In the chromatogram corresponding to the fifth injection, two chromatographic peaks (isomer 1 and 2 [6] at retention times 24.3 and 25.3 min, respectively) appeared and stayed present in the next chromatograms, up to chromatogram 16. The problem reoccurred again from 17th to 20th injection. In Table 2, results from the repeatability study for cycloserine dimer are summarized. Based on the RSD results shown in Table 2, the Ph. Int. HPLC method was not found to be repeatable for the determination of the cycloserine dimer. Therefore, further verification of the Ph. Int. HPLC method was skipped.

When designing a new HPLC quantification study of d-cycloserine and its related compounds, one should be careful, as d-cycloserine might not be stable in the solvent used (mobile phase A; see Section 3.3.2), since it contains acetonitrile and a buffer with a pH of 2.8. According to the literature, dimerization of d-cycloserine was initiated by acetonitrile as a sample solvent, therefore, promoting methanol as a solvent of choice [6,21]. Conversion of d-cycloserine to the cycloserine dimer was found to be further enhanced in the ESI source of the MS [21]. Moreover, d-cycloserine was unstable under acidic conditions [12,22], and converted to its dimer in the solid-state and in solution [22]. The dimer was also shown to exist in a pH-dependent equilibrium (pH 1-2) with cycloserine [22]. The addition of citrate or ethylenediaminetetraacetic acid into the mobile phase might prevent the potential on-column dimerization of d-cycloserine, as reported by Heaton et al. (2019) [24]. Therefore, a future study, where the sample solvent and the pH are carefully controlled might solve some of the issues with the quantification of d-cycloserine and its related substances in pharmaceutical dosage forms. Furthermore, a systematic method development and thorough troubleshooting approach are needed to reveal the right reasons for the unrepeatability of d-cycloserine dimer areas. This would aid the development of a reliable HPLC–UV method for the determination of d-cycloserine-related compounds, thus, increasing the safety of the produced d-cycloserine drugs. However, development of new HPLC method was out of the scope of this manuscript and alternative NMR method for the determination of d-cycloserine impurities in final dosage forms, was examined.

### 2.2. Development and Verification of the ^1^H DOSY NMR Method for Determination of Impurities in Samples Containing d-Cycloserine

The HPLC method performed according to The International Pharmacopoeia failed to show repeatability of the cycloserine dimer chromatographic peak areas and, therefore, was not considered to be suitable for quantification of the cycloserine dimer. Therefore, a ^1^H DOSY NMR method was developed as an alternative method for the determination of d-cycloserine impurities in its formulations.

However, NMR spectroscopy was less sensitive with respect to the HPLC method. Therefore, the limit of detection (LOD) and LOQ were determined for the cycloserine dimer, with initial NMR experiments. Table 3 presents the *S*/*N* ratio of selected NMR signals at different concentrations of the cycloserine dimer in the ^1^H and ^1^H DOSY NMR spectra. Impurity profiling methods should achieve *S*/*N* ≥ 10, according to The International Pharmacopoeia, which was satisfied at 4 μg mL^−1^ for ^1^H NMR and 10 or 20 μg mL^−1^ for the DOSY NMR spectrum (Table 3, NMR spectra of the cycloserine dimer are shown in the Appendix A).

The established HPLC method in The International Pharmacopoeia [10,11] asks for the preparation of capsule solution containing 0.5 mg mL^−1^ of d-cycloserine. For the ^1^H DOSY NMR method, concentrations of d-cycloserine and impurity solutions were increased due to the lower sensitivity of NMR. The high solubility of d-cycloserine in water allowed preparation of samples, where concentration of the d-cycloserine solution was 15 mg mL^−1^, and the specification limit for impurities was achieved at 60 μg mL^−1^ for the final dosage form (0.4%), and at 22.5 μg mL^−1^ (0.15%) for API (Active Pharmaceutical Ingredient). Signals of substances at these concentrations gave the appropriate *S*/*N* ratio for the DOSY NMR technique. Figure 4 shows the ^1^H DOSY NMR spectrum of the 0.15% cycloserine dimer, relative to the concentration of d-cycloserine in the sample solution. Well-resolved NMR signals of the d-cycloserine and cycloserine dimer allowed the determination of the translational diffusion coefficient for API and dimer impurity, with the values of 5.5 × 10^−10^ m^2^ s^−1^ and 3.2 × 10^−10^ m^2^ s^−1^ for the d-cycloserine and the cycloserine dimer, respectively.

In addition to the d-cycloserine and cycloserine dimer signals, NMR signals of the unidentified impurities were observed in the ^1^H spectrum, which most likely corresponded to the degradation products or process-related impurities of the d-cycloserine, as they were also observed in the ^1^H NMR spectra of the pure active substance d-cycloserine.

Additionally, 0.05% cycloserine dimer, relative to the concentration of d-cycloserine in the sample solution was prepared and analyzed to reach the quantification limit. The ^1^H NMR and ^1^H DOSY NMR spectra are shown in the Appendix A, and once again the NMR signals of the d-cycloserine and cycloserine dimer were well-resolved.

Two other impurities were potentially found in the d-cycloserine products—3-chloro-d-alanine methyl ester hydrochloride and d-serine. Both impurities are process impurities of d-cycloserine, whereas the second is also the decomposition product of the d-cycloserine final dosage form. The method for determining the impurity profile must allow sufficient resolution between the peaks or otherwise enable proper identification and quantification of impurities. The ^1^H NMR spectra of all three impurities and the d-cycloserine are shown in Appendix A.

Quantitative NMR analysis was performed for the sample solution containing all three impurities and d-cycloserine. Well-resolved ^1^H NMR signals of each compound were used (Figure 5), with the addition of internal standard maleic acid, to determine the content of each compound in the mixture according to well-known equation:(1)$$P_{x}=\frac{I_{x}\times N_{RS}\times M_{x}\times P_{RS}\times m_{RS}}{I_{RS}\times N_{x}\times M_{RS}\times m_{x}}$$
where *m_x_* is the (calculated) mass of the analyte; *m_RS_* is the mass of the reference substance; *I_x_* is the integral of the analyte signal; *I_RS_* is the integral of the reference substance (= 2.000); *N_x_* is the number of analyte proton nuclei; *N_RS_* is the number of reference substance proton nuclei (= 2); *M_x_* is the molecular mass of the analyte; *M_RS_* is the molecular mass of the reference substance (= 116.07 gmol^−1^); *P_x_* is the purity of the analyte as mass fraction; and *P_RS_* is the purity of the reference substance as mass fraction.

The accuracy and reliability of the ^1^H NMR quantification was established for the d-cycloserine and its impurities, in comparable concentrations. We demonstrated that measurements of the intended analytes’ signals were free of interference from the reference substance, as well as from impurities (Figure 5). The values of the integrals were reported relative to the signal of maleic acid, which was arbitrarily set at a value of 2. The calculated and actual contents of d-cycloserine and its impurities in the sample solution are given in Table 4. It was shown by Malz and Jancke that single pulse ^1^H NMR fulfils all requirements to be used as a validated method for quantitative determinations of the amount fractions of dissolved sample mixtures, where all issues regarding linearity, robustness, specificity, selectivity, and accuracy were considered [33]. On the other hand, the NMR stability test (Appendix A) and T_1_ relaxation times (Appendix A) for each compound were determined and are reported in the Appendix A.

The targeted NMR method should be able to characterize and quantify d-cycloserine and its impurities at the specification limit (NMT 0.15% with respect to concentration of d-cycloserine in API and NMT 0.4% with respect to concentration of d-cycloserine in final dosage form [10,11]). However, to employ the DOSY technique for characterization based on different molecular weight, one has to meet the following criterion [25]:(2)$$\frac{M_{Wa}-M_{Wb}}{{\bar{M}}_{W}}\;\geq 0.10$$

Certain pairs of compounds satisfy this condition easily (d-cycloserine − cycloserine dimer = 0.67), whereas others fail to do so (e.g., d-serine − d-cycloserine = 0.03). Presence of all three impurities in a final dosage form is the worst-case scenario and most likely will not occur. In addition, it can be controlled, as the 3-chloro-d-alanine methyl ester hydrochloride is a process impurity that can be limited below the specification limit, during the synthesis of pure active substance d-cycloserine. As a result, a ^1^H NMR spectrum of high resolution and acceptable selectivity can be obtained (Appendix A).

The suitability of the method was eventually evaluated on a real sample. d-cycloserine capsule solution was prepared to obtain d-cycloserine concentration of about 5 mg mL^−1^. Specification limit for each impurity was 20 μg mL^−1^, which satisfied the S/N requirement. Analysis of sample solutions through NMR can be performed for the samples, where at least 80% of API is still intact. The stress stability procedure usually results in 90% of intact API. Therefore, the d-cycloserine capsules packed in a primary container were exposed to accelerated stress conditions (at 40 °C, RH 30%) for 10 days, which resulted in 87% of intact API, as established by the ^1^H NMR spectrum. The ^1^H DOSY NMR spectrum of the d-cycloserine capsule solution did not show the presence of process impurity (3-chloro-d-alanine methyl ester hydrochloride, Figure 6). Therefore, DOSY allowed us to distinguish NMR signals attributed to API, its key degradation products, and related substances. NMR assays were calculated based on the ^1^H NMR spectra. The ^1^H NMR spectra were used to establish a concentration of cycloserine dimer (0.1 mg mL^−1^) in a sample solution, which corresponded to 2% of d-cycloserine and was above the specification limit (Appendix A).

## 3. Materials and Methods

### 3.1. Chemicals

Acetonitrile (ACN) HPLC gradient grade was purchased from Fisher Scientific (Leicester, UK). Sodium 1-octanesulfonate (SOS) was acquired from Sigma-Aldrich (St. Louis, MO, USA), while potassium dihydrogen phosphate (KH_2_PO_4_), and phosphoric acid (H_3_PO_4_; 85%, analytical grade) were purchased from Merck (Darmstadt, Germany). Demineralized water was supplied by the Milli-Q water purification system (Millipore, Milford, MA, USA).

d-cycloserine (99.6%; USP standard) was used as a standard for the active pharmaceutical ingredient. Other standards used in the study were d-serine (100%), which was purchased from Sigma-Aldrich, cycloserine dimer (3,6-bis(aminooxymethyl)piperazine-2,5-dione) (95%) from Toronto Research Chemicals Inc. (North York, ON, Canada), and 3-chloro-d-alanine methyl ester hydrochloride (97.3%) from Sigma-Aldrich (St. Louis, MO, USA).

d-cycloserine capsules were manufactured by The Chao Center (West Lafayette, IN, USA).

For the NMR experiments, 99.9% deuterated water from Merck was used as a solvent. Maleic acid (99.98%; TraceCERT^®^ CRM) was purchased from Sigma-Aldrich (St. Louis, MO, USA). Deuterated sodium 3-(trimethylsilyl)propionate (TMSP) was purchased from ARMAR Chemicals (Döttingen, Switzerland).

### 3.2. Instrumentation

HPLC chromatograms were acquired using the Waters 2695 Alliance liquid chromatographic system (Milford, MA, USA), equipped with the UV Waters 2996 photodiode array detector (Milford, MA, USA). Empower3 software (Waters, Milford, MA, USA) was used to record and process the signals.

1D ^1^H and ^1^H-DOSY NMR spectra were collected on an Agilent Technologies DD2 600 MHz NMR spectrometer (Santa Clara, CA, USA), equipped with the HCN Cold probe.

### 3.3. Verification of The International Pharmacopoeia (Ph. Int.) HPLC Method for Determination of Impurities in Samples Containing d-Cycloserine

#### 3.3.1. Chromatographic Conditions

Potassium phosphate monobasic buffer (0.2 M) was prepared by dissolving 27.2 g of KH_2_PO_4_ in 800 mL water, adjusting the pH at 2.8 ± 0.05 with 85% H_3_PO_4_ and adding water to the total volume of 1000 mL. SOS solution (20 mM) was prepared by dissolving 4.7 g SOS in 1000 mL water. Mobile phases A and B, both consisting of ACN, 20 mM SOS, 0.2 M KH_2_PO_4_ buffer at pH 2.8 and water were prepared as specified in the Ph. Int. in the following ratios—mobile phase A: ACN/SOS/KH_2_PO_4_ buffer/water = 4/70/10/16 *v*/*v*/*v*/*v*; and mobile phase B: ACN/SOS/KH_2_PO_4_ buffer/water = 17/70/10/3 *v*/*v*/*v*/*v*.

A Hypersil BDS C18 HPLC column (250 × 4.6 mm; 5 µm) from Thermo Fischer Scientific (San Jose, CA, USA) was used for the analysis. Gradient elution was applied at 45 °C and a flow rate of 1 mL min^−1^: 0% B (0–16 min), 0%–100%B (16–18 min), 100% B (18–22 min), 100%–0%B (22–24 min), and 0% B (24–30 min). The detection wavelength was set at 219 nm and the injection volume was 50 µL.

#### 3.3.2. Standard and Sample Preparation

d-cycloserine capsule content, corresponding to 50 mg of the active compound was weighted and dissolved in 100 mL mobile phase A, to obtain a solution with a concentration of 0.5 mg mL^−1^. This solution was mixed and filtered through 0.45 µm Millipore Millex HV, PVDF filter, Merck.

A standard stock solution of d-cycloserine was prepared in a concentration of 0.5 mg mL^−1^ in mobile phase A (50 mg d-cycloserine in 100 mL mobile phase A). This solution was further diluted to obtain a standard solution of 0.1% d-cycloserine, relative to its concentration in the sample (0.1 mL from the standard stock solution was transferred to a 100 mL flask and mobile phase A was added to the mark).

To check the repeatability of d-cycloserine chromatographic peak areas, 10 mg of the d-cycloserine standard was dissolved in 50 mL mobile phase A and was ultrasonicated for 5 min until complete dissolution. It was further diluted (0.75 mL from the stock solution was transferred to 100 mL flask and mobile phase A was added to the mark) to obtain a solution with a concentration of 0.0015 mg mL^−1^.

To check the linearity and limit of quantification (LOQ), 2 mg of the d-cycloserine standard were dissolved in 10 mL mobile phase A, and ultrasonicated for 5 min until complete dissolution. The solution was further diluted (2 mL from the stock solution was transferred to a 20 mL flask and mobile phase A was added to the mark), to obtain a working solution (WS). The working solution was again diluted for LOQ and linearity studies, as shown in Table 1.

To check the retention times of d-cycloserine, 3-chloro-d-alanine methyl ester hydrochloride, and d-serine, 7.7 mg of each standard was dissolved in 10 mL of mobile phase A and ultrasonicated for 5 min until complete solvation, to obtain concentrations of about 0.77 mg mL^−1^.

For column conditioning with cycloserine dimer and repeatability check of the cycloserine dimer chromatographic peak areas, the cycloserine dimer standard was prepared as the previous three standards in mobile phase A, with a concentration of about 0.77 mg mL^−1^.

To acquire UV spectra, separate solutions containing d-cycloserine, 3-chloro-d-alanine methyl ester hydrochloride, and d-serine with concentrations of about 0.02 mg mL^−1^ were prepared. Each standard (2 mg) was dissolved in 70 mL mobile phase A, and the solution was ultrasonicated for 5 min, until complete solvation and finally filled up to the volume (100 mL) with mobile phase A. UV spectra were acquired in the range from 190 nm to 400 nm, using a quartz cuvette. A spectrum of the blank containing only mobile phase A, was also acquired.

### 3.4. Development and Verification of the ^1^H DOSY NMR Method for Determination of Impurities in Samples Containing d-Cycloserine

Since the active pharmaceutical ingredient, d-cycloserine, as well as all impurities were well-soluble in water, the samples were dissolved in D_2_O, using ultrasonic bath to obtain concentrations of 2, 4, 10, and 20 μg mL^−1^. To each sample, the TMSP salt was added as the internal standard for a relative chemical shift and maleic acid, for quantification.

Diffusion coefficients were collected using pulsed-field gradient echo pulse sequence. The acquisition time for the ^1^H and ^1^H DOSY spectra were 4.0 s, and the recorded spectra were the result of 64 scans. Recycle delay of 10 s was used for the ^1^H spectra. The chemical shifts were given in parts per million (ppm), relative to the TMSP salt at 0 ppm.

## 4. Conclusions

The Ph. Int. HPLC method for the d-cycloserine capsule impurity profiling showed good repeatability, linearity, and *S*/*N* ratio for d-cycloserine. However, the method failed to show the repeatability of the chromatographic peak areas corresponding to the cycloserine dimer—one of the known d-cycloserine impurities—and was, therefore, not considered to be suitable for its quantification. Thus, the DOSY NMR method for determination of d-cycloserine impurities in its formulations was developed. Specification limit of d-cycloserine was achieved at 22.5 μg mL^−1^, which gave an appropriate *S*/*N* ratio for the DOSY NMR technique. Well-resolved NMR signals of the d-cycloserine and cycloserine dimer allowed for the determination of translational diffusion coefficient for API and dimer impurity, with the values 5.5 × 10^−10^ m^2^ s^−1^ and 3.2 × 10^−10^ m^2^ s^−1^ for d-cycloserine and cycloserine dimer, respectively. Furthermore, the accuracy and reliability of the ^1^H NMR quantification was established for d-cycloserine and its impurities, in comparable concentrations. We demonstrated that measurements of the intended analytes’ signals were free of interference from the reference substance, as well as from impurities. Suitability of the NMR method was eventually evaluated on a real sample. d-cycloserine capsules packed in the primary container were exposed to accelerated stress conditions (10 days at 40 °C, RH 30%). DOSY allowed us to distinguish NMR signals attributed to API and its degradation products. The NMR assay was calculated, based on the ^1^H NMR spectra. The ^1^H NMR spectra were used to establish the concentration of the cycloserine dimer, which corresponded to 2% of d-cycloserine and was above the specification limit.

The method described in this study, which combined the ^1^H and DOSY NMR techniques, offers a new approach for practical use in analytical laboratories of the pharmaceutical industry. DOSY NMR has been found to be particularly useful due to its ability to perform virtual separation of the d-cycloserine and cycloserine dimer impurity, without engaging in a physical separation scheme for the purification of each components. A similar approach could be used for other compounds, as long as the specificity and selectivity were achieved for both ^1^H and DOSY NMR.

## Figures and Tables

**Figure 1 molecules-25-01684-f001:**
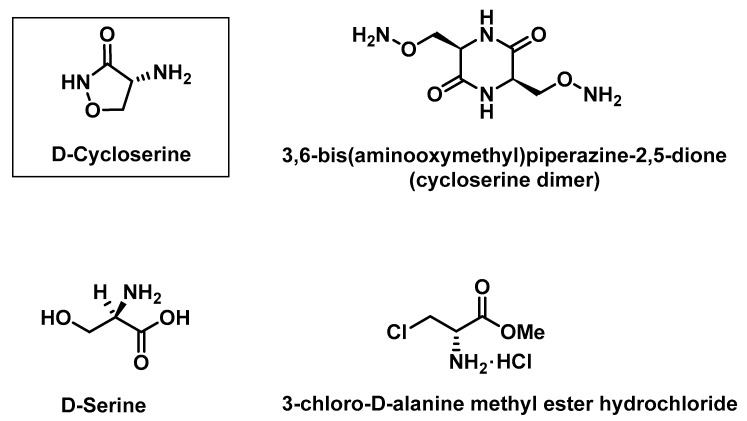
Chemical structure of the d-cycloserine and its impurities.

**Figure 2 molecules-25-01684-f002:**
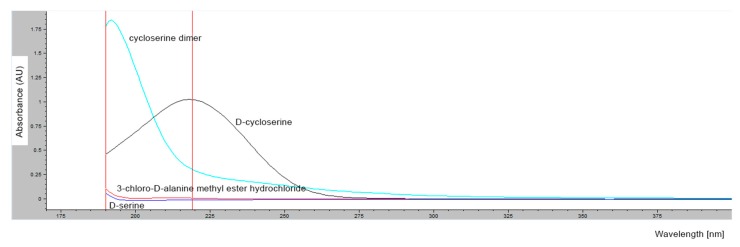
UV spectra of d-cycloserine, d-serine, cycloserine dimer, and 3-chloro-d-alanine methyl ester hydrochloride.

**Figure 3 molecules-25-01684-f003:**
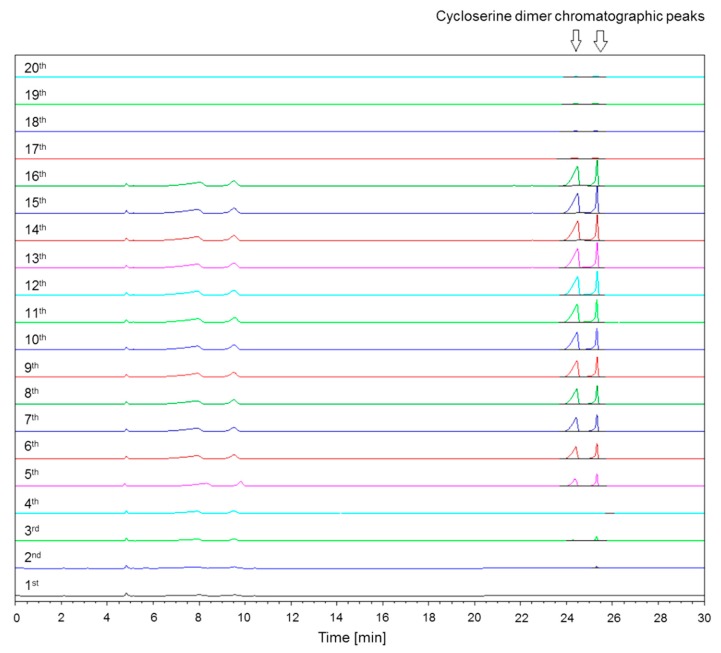
The chromatograms of twenty injections of the cycloserine dimer in sequential injection order.

**Figure 4 molecules-25-01684-f004:**
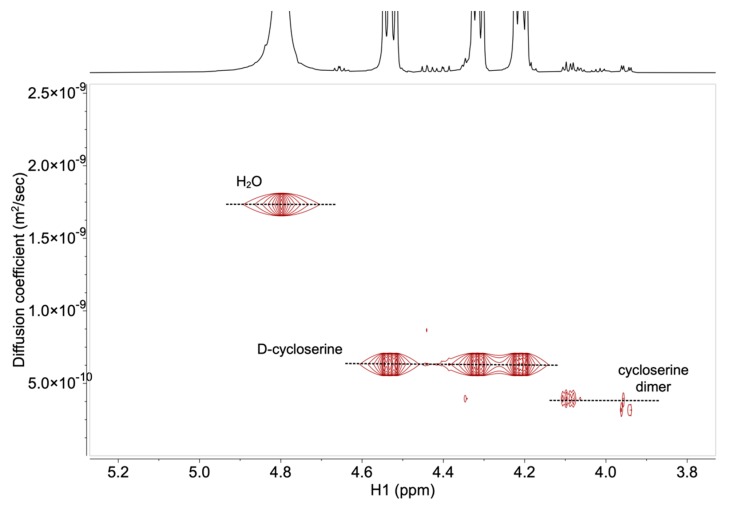
^1^H NMR (top) and ^1^H DOSY NMR spectra (bottom) of the d-cycloserine and cycloserine dimer within the specification limit (0.15% relative to the concentration of d-cycloserine in the sample solution). The diffusion coefficient (D) is reported in m^2^ s^−1^ × 10^−10^.

**Figure 5 molecules-25-01684-f005:**
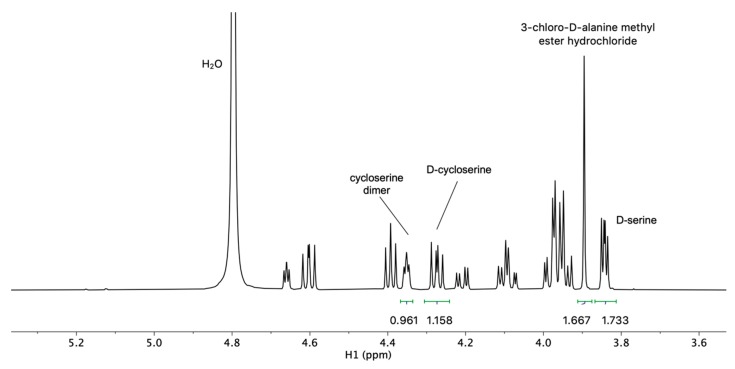
^1^H NMR spectrum of d-cycloserine and three impurities 3-chloro-d-alanine methyl ester hydrochloride, d-serine, and cycloserine dimer, together with the integral values for each of the signals that was used to calculate the content of each component (replicate 2).

**Figure 6 molecules-25-01684-f006:**
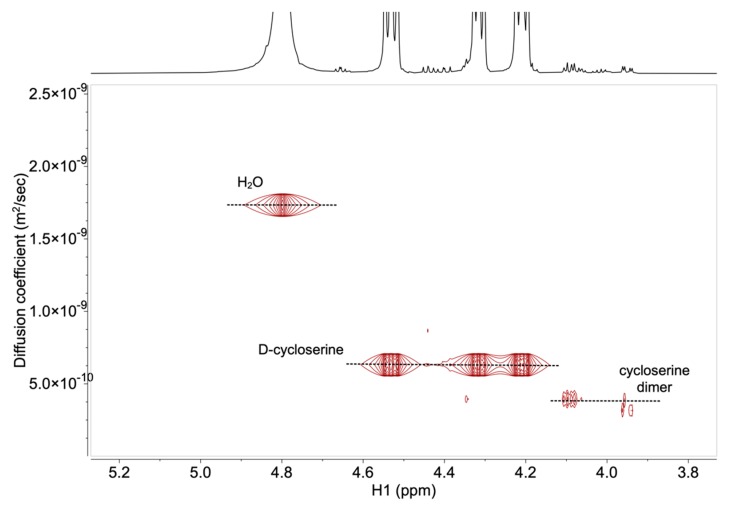
The ^1^H DOSY NMR spectrum of the d-cycloserine capsule solution exposed to accelerated stress conditions. Diffusion coefficient (D) was reported in m^2^ s^−1^ × 10^−10^.

**Table 1 molecules-25-01684-t001:** Dilution protocol to obtain the solutions for the limit of quantification (LOQ) and linearity studies.

Solution Name	Volume WS [mL]	% d-Cycloserine ^1^	Concentration [µg mL^−1^]
LOQ1	0.005	0.001	0.005
LOQ2	0.05	0.01	0.05
LOQ3	0.10	0.02	0.10
LOQ4	0.15	0.03	0.15
LOQ5 = Lin1	0.25	0.05	0.25
Lin2	0.30	0.06	0.30
Lin3	0.35	0.07	0.35
Lin4	1.0	0.2	1.0
Lin5	1.2	0.24	1.2
Lin6	1.5	0.3	1.5
Lin7	2.0	0.4	2.0
Lin8	2.4	0.48	2.4
Lin9	2.65	0.53	2.65

^1^ Relative to its concentration in the sample.

**Table 2 molecules-25-01684-t002:** Repeatability of the cycloserine dimer chromatographic peak areas.

Injection	Area [µV*sec] Cycloserine Dimer 1	Area [µV*sec] Cycloserine Dimer 2	Total Area [µV*sec] Cycloserine Dimers 1 & 2
1	0	32794	32794
2	0	136069	136069
3	94450	435062	529512
4	0	0	0
5	1571131	1261973	2833104
6	2813514	158198	2971712
7	3532956	1786999	5319955
8	4015227	1960635	5975862
9	4402109	2117970	6520079
10	4723430	2260472	6983902
11	4995383	2392454	7387837
12	5222891	2516870	7739761
13	5419184	2634801	8053985
14	5593956	2748004	8341960
15	5745123	2855549	8600672
16	5509693	2779019	8288712
17	136259	151090	287349
18	100282	132602	232884
19	79771	124348	204119
20	65357	119038	184395
RSD (%) (1–20):	90.0	87.7	88.5
RSD (%) (8–20):	68.9	65.8	67.8
RSD (%) (8–16):	11.6	12.7	11.9

**Table 3 molecules-25-01684-t003:** *S/N* ratios at different concentrations of cycloserine in the ^1^H and ^1^H DOSY NMR spectra.

	Concentration [μg mL^−1^]	*S/N* at δ 4.4 ppm	*S/N* at δ 4.1 ppm	*S/N* at δ 4.0 ppm
^1^H	20	39.1	42.4	63.2
	10	17.5	26.1	30.9
	4	11.3	13.1	12.3
	2	9.9	7.7	7.7
^1^H DOSY ^1^	20	12.5	20.7	23.9
	10	6.6	11.0	16.4

^1^ First increment of the magnetic field gradient in the DOSY NMR spectrum.

**Table 4 molecules-25-01684-t004:** Calculated vs. actual content of d-cycloserine and its impurities in the sample solution ^1^.

Compound	Replicate	m_x_	I_x_	N_x_	m_RS_	Calculated P_x_ (%)	Actual P_x_ (%)
d-cycloserine ^1^	replicate 1	2.908	1.365	1	2.256	93.122	99.6
	replicate 2	1.954	1.158	1	1.788	93.181	99.6
	replicate 3	2.084	0.909	1	2.445	93.782	99.6
					**average:**	**93.4**	**99.6**
Cycloserine dimer ^2^	replicate 1	1.447	0.712	2	2.256	97.627	95.0
	replicate 2	1.581	0.961	2	1.788	95.582	95.0
	replicate 3	1.584	0.714	2	2.445	96.926	95.0
					**average:**	**96.7**	**95.0**
3-chloro-d-alanine methyl ester HCl ^3^	replicate 1	1.491	1.255	3	2.256	94.886	97.3
	replicate 2	1.560	1.667	3	1.788	95.472	97.3
	replicate 3	1.988	1.610	3	2.445	98.943	97.3
					**average:**	**96.4**	**97.3**
d-serine ^4^	replicate 1	1.557	0.810	1	2.256	106.240	100.0
	replicate 2	2.728	1.733	1	1.788	102.820	100.0
	replicate 3	3.531	1.631	1	2.445	102.233	100.0
					**average:**	**103.8**	**100.0**

^1^ M(d-cycloserine) = 102.09 gmol^−1^. ^2^ M(dimer) = 204.20 gmol^−1^. ^3^ M(ester) = 174.03 gmol^−1^. ^4^ M(d-serine) = 105.09 gmol^−1^; For all replicates: M_RS_ = 116.07 gmol^−1^, P_RS_ = 99.98, I_RS_ = 2.000, and N_RS_ = 2.

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
