# Peer review of "Determination of d-Cycloserine Impurities in Pharmaceutical Dosage Forms: Comparison of the International Pharmacopoeia HPLC–UV Method and the DOSY NMR Method"

_molecules, 2020, doi:10.3390/molecules25071684_

Round 1
Reviewer 1 Report
In this contribution, Makuc et al. present an analytical work demostrating the the Pharmacopeia HPLC-UV method for determining D-cycloserine impurity profiling is not repeatable. After a careful analysis of the existing methodology, the variable response of cycloserine dimer (one of D-cycloserine impurities) was clearly determined. In order to improve this fail, the DOSY NMR technique is further analysed to determine its capabilities for measure the levels of D-cycloserine impurities in pharmaceutical dosage. After a detailed work, the authors demonstrated that DOSY NMR method allows accurate identification and quantification of cycloserine dimer and can be used in analytical laboratories of the pharmaceutical industry.
The article is well presented and clear. All the methodology applied is presented with details and the motivation for each task is well explained and documented. Thus, in my opinion, the manuscript is of interest for the audience of Molecule, specially for those interested in analytical chemistry methods.
The only minor suggestion to the author is to revise the final sentence in the Introduction section. The sentence "Few reports exist in the literature..." is confusing since it refer to different systems and techniques (19F NMR) that does not closely linked to the cycloserine issue.
Reviewer 2 Report
The authors of the manuscript have found inconsistencies in the determination of D-cycloserine in pharmaceutical capsules when applying the method of the International Pharmacopoeia. HPLC or UPLC coupled to tQ-MS detectors should be the method of choice to solve this analytical problem when impurities are at low levels. But the literature describes the formation of artifacts, such dimers, in the ionization source or, even, in the mobile phase solution. Given these facts, the authors propose the use of 1H-DOSY experiments. Detection and quantification levels between NMR and HPLC-MS do not allow for comparison, but the authors found sufficient sensitivity using DOSY NMR experiments to achieve quality specifications. The experiments with HPLC-UV carried out and those according to the method of the International Pharmacopoeia, together with the experiments with NMR seem consistent and with ruggedness, so I recommend its publication, with a minor revision, in the journal Molecules.
Consideration.
Lines 136-138:
In HPLC-UV analysis of the cycloserine dimer, two peaks appear, but using HPLC-MS, I understand exactly the same conditions, only one appears. No reason has been given for this fact, and I understand that if the main analytical problem with the use of HPLC is the formation of artifacts, an adequate description of this phenomenon has been given.
Reviewer 3 Report
D-cycloserin play important role in many diseases and its impurities can affect on its therapeutic parameters. Chromatographic method was not considered as suitable for quantification of cycloserine dimer. This manuscript described convenient method to analyze the D-cycloserine and its impurities. The Authors have showed the 1H DOSY method allowed to distinguish NMR signals attributed to D-cycloserine, its dimer and other products.
I think that this manuscript is easy to understand but before the manuscript will accepted for publication, the authors should explain all the used abbreviations (for example: API).
Can this method be used for other compounds? Which?
